# Simultaneous invasion decouples zebra mussels and water clarity

Heidi M. Rantala [1✉], Donn K. Branstrator[2], Jodene K. Hirsch[1], Thomas S. Jones [1] & Gary Montz[1]

Species invasions are a leading threat to ecosystems globally, but our understanding of interactions among multiple invasive species and their outcomes on ecosystem properties is undeveloped despite their significance to conservation and management. Here we studied a large lake in Minnesota, USA, that experienced a simultaneous surge in invasive zebra mussel and spiny water flea populations. A long-term (2000–2018) dataset offered a rare opportunity to assess whole-ecosystem shifts following the co-invasion. Within two years, the native crustacean zooplankton community declined abruptly in density and productivity (−93% and −91%, respectively). Summer phytoplankton abundance and water clarity remained stable across the time series, an unexpected outcome given the high density of zebra mussels in the lake. Observational data and modeling indicate that removal of native herbivorous zooplankton by the predatory spiny water flea reduced zooplankton grazing pressure enough to compensate new grazing losses due to zebra mussels, resulting in a zero net effect on phytoplankton abundance and water clarity despite a wholesale shift in secondary production from the pelagic to the benthic food web. This study reveals the extent of direct and indirect effects of two aquatic invaders on food-web processes that cancel shifts in water clarity, a highly valued ecosystem service.

[1] Minnesota Department of Natural Resources, St. Paul, MN, USA. [2] Department of Biology, University of Minnesota Duluth, Duluth, MN, USA. ✉email: heidi.rantala@state.mn.us

Invasive species have changed a vast number of Earth's ecosystems and caused far reaching environmental and economic damage[1]. In the United States alone, annual costs associated with invasive species are almost $120 billion[2]. After decades of research, scientific understanding of the outcomes of single species invasions on ecosystem properties is strong in many cases and offers a sound framework for prediction and management. However, understanding of the impacts of two or more invasive species that co-occur in an ecosystem is still maturing. On one hand, there is growing evidence for the invasional meltdown hypothesis whereby multi-species assemblages of nonindigenous taxa facilitate one another's invasion and impacts[3,4]. On the other hand, the predicted direction of those impacts are less certain and may be additive (summation), synergistic (amplification), or antagonistic (mitigation)[5,6]. The complexity of multi-species invasion scenarios is further compounded by nonlinearity in the relationships between species density and impacts, lag times between arrival and population growth, and priority effects[7–9]. These gaps warrant further attention given the stark inevitability that continued homogenization of Earth's biota will surely accelerate the number of ecosystems that support multiple invasive species[10,11].

The zebra mussel (*Dreissena polymorpha*, hereafter *Dreissena*) and spiny water flea (*Bythotrephes cederströmii*, hereafter *Bythotrephes*) are two of North America's most widespread and problematic aquatic invasive species[12,13]. When new populations are established, both species can create large environmental impacts including competition with native species for food and habitat, predation on native species at multiple trophic levels, biophysical and biochemical alterations of benthic habitat, and changes in water clarity and nutrient pathways[14–18]. Both invaders reached high densities (*Bythotrephes*, median = 6.8 individuals m$^{-3}$ in 2010, *Dreissena*, 9536 individuals m$^{-2}$ in 2011) within one year of each other in 2010–2011 in Lake Mille Lacs, a large, shallow, polymictic, mesotrophic lake (surface area = 520 km$^2$, mean depth = 8.7 m, maximum depth = 12.8 m) in central Minnesota, USA, offering a rare opportunity to assess the outcomes of a co-invasion (Figs. 1 and 2). While the lake periodically stratifies during calm conditions, the geometry of the lake (geometry ratio = 11.9 m$^{-0.5}$)[19] makes stratification unstable and short-lived, typical of seasonally polymictic lakes (Fig. 2)[20,21].

*Dreissena* is native to the Ponto-Caspian basin and has been a dominant invader throughout North America since the 1980s. Its rapid filtering capacity and extraordinary densities[22] have led to predictable reductions in phytoplankton biomass (about 50%) and zooplankton biomass (about 40–60%) and increases in water clarity (about 35–50%) across invaded food webs, as well as wholesale translocations of nutrients and energy from the water column to the benthos[23–25]. The enormous threat of *Dreissena* to freshwater lakes and rivers has led to widespread and ongoing international efforts to study, contain, and eradicate the species.

*Bythotrephes* is an aggressive Eurasian invader that also was first recorded in North America in the 1980s and is widely distributed throughout the Laurentian Great Lakes region. *Bythotrephes* preys on native herbivorous zooplankton with the potential to consume more herbivorous zooplankton than produced seasonally[26–28] and more than consumed by planktivorous fish and native predatory zooplankton in localized regions[29–31]. *Bythotrephes* is credited with indirectly decreasing lake water clarity by consuming native zooplankton grazers and activating a trophic cascade[32], although its effects on phytoplankton are mixed[33–35], with algae both increasing and not changing in invaded lakes. While it may be predictable that a co-invasion of *Dreissena* and *Bythotrephes* could force opposite effects on water clarity based on first principles of top-down control, this hypothesis has not been tested at the whole-ecosystem scale.

Here we document the response of Lake Mille Lacs to co-invasion by *Dreissena* and *Bythotrephes* as part of an annual (monthly from May to September), long-term (2006–2018), whole lake (Fig. 1) observational monitoring program of the food web (phytoplankton, zooplankton, and fishes). We examined temporal trends in water clarity using a hierarchical generalized additive model, including a variable for invasion status based on the first detection of both *Dreissena* and *Bythotrephes* (Fig. 3). We modeled grazing rates of *Dreissena* and herbivorous zooplankton as well as zooplankton secondary production that included body

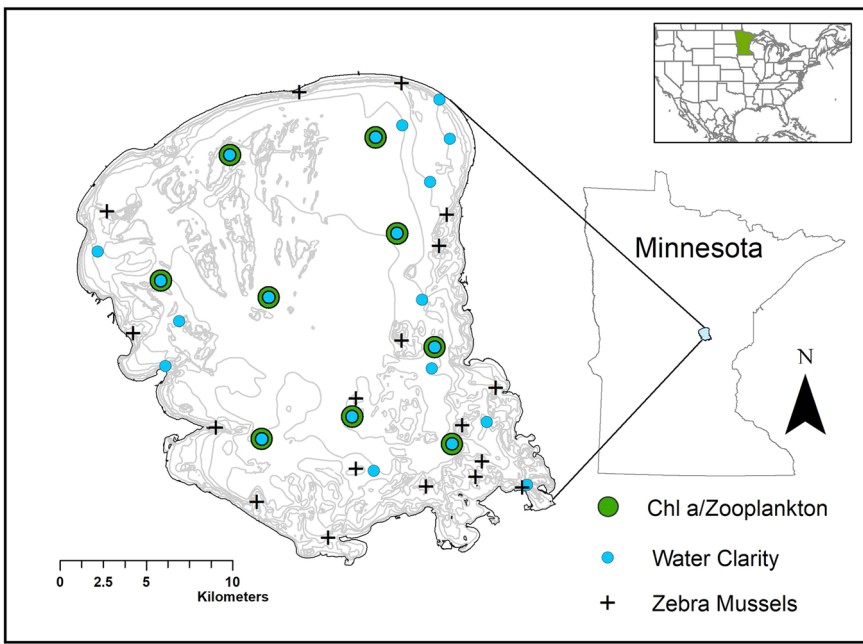

**Fig. 1 Map of Lake Mille Lacs[78,79].** Nine zooplankton/chlorophyll *a* sampling stations (green circles), 18 zebra mussel transect locations (black crosses), and 21 water clarity stations (blue circles) were widely distributed across the lake. The depth contour lines (gray) are at 1.5 m intervals. Inset: Location of the state of Minnesota in the United States[88,89].

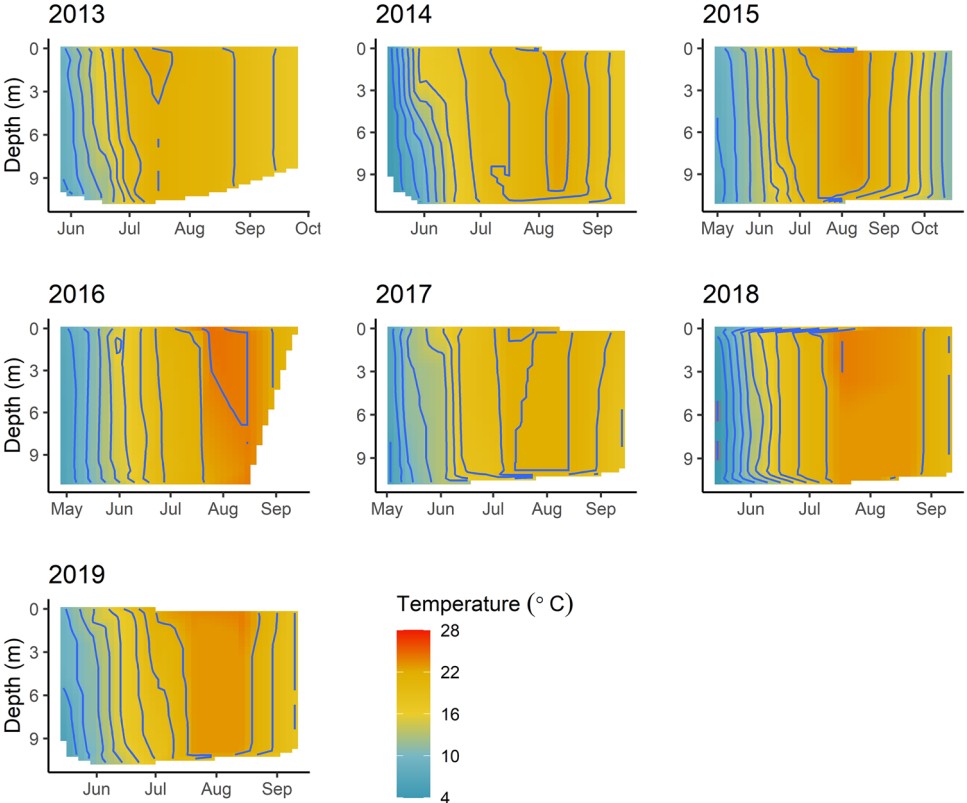

**Fig. 2 Temperature (°C) heat maps from May-September/October of 2013-2019 from Lake Mille Lacs.** The heat maps were produced by extrapolating temperature profile data between sampling dates. We excluded years prior to 2013 due to incomplete data.

length measurements of about 94,000 individual zooplankton and the enumeration of about 616,000 individual *Dreissena* over the time series. Secondary production is the formation of heterotrophic biomass through time and integrates growth, reproduction, and survivorship, making it a holistic population metric[36]. Taxon-specific grazing rates are measures of the ability of animals to influence water clarity and affect nutrient flow, making it possible to compare the functional roles of native grazing zooplankton to *Dreissena*.

## Results and discussion

**Water clarity**. Water clarity was measured at 21 stations ($n = 1224$ total clarity measurements) at a frequency of 12-114 (median = 59) observations per year (Fig. 1). It ranged between 1.4 and 7.9 m across the study period. In early summer (May and June, days 123–180, Fig. 3, Supplementary Fig. 1) water clarity varied among the three invasion status categories (prior to *Dreissena* or *Bythotrephes* invasion (2000–2005), post-*Dreissena* invasion (2006–2009), and post-*Dreissena* and *Bythotrephes* invasion (2010–2018)). Through the remainder of the summer into early fall (July—September, days 181-269) water clarity was not different between the pre-invasion and duel invasion scenarios, but summer water clarity was higher in 2006–2009 when the *Dreissena* population was expanding prior to *Bythotrephes* invasion (Fig. 3). There was no evidence of spatial patterns in the clarity data to suggest the impacts of *Dreissena* were localized (Supplementary Fig. 2) near the mussel beds or nearshore.

**Invasive species populations**. *Dreissena* abundance ranged from a total of four detected individuals in 2005 to maximum densities of ~14,000 individuals m$^{-2}$ in 2012, demonstrating a typical lag phase in population growth (Fig. 4A)[37,38]. After *Dreissena*

densities peaked, they decreased gradually to half their peak density by 2017. In other studies, water clarity usually increased after invasion by *Dreissena*[24,39], although this change was sometimes buffered by large sediment loads or cyanobacteria blooms driven by increased phosphorus recycling by *Dreissena*[40,41]. In the western basin of Lake Erie, which is thermally mixed, *Dreissena* cleared 20% of the water column daily[42], and in a meta-analysis of 13 north-temperate, thermally mixed lakes, chlorophyll *a* declined by ~40% post-invasion of *Dreissena* spp.[43]. In contrast, although *Dreissena* cleared 15-32% of the water column daily in Lake Mille Lacs between 2011 and 2018 (Fig. 4G, Lake Mille Lacs volume = 4.83 km$^3$) we did not observe a trend of decreasing chlorophyll *a* through time (Fig. 4F). We observed increased clarity in spring but not summer or early autumn (Fig. 3) and no overall change averaged across spring to autumn (Fig. 4F).

*Bythotrephes* abundance ranged from a total of two detected individuals in September 2009 to maximum densities of ~15 individuals m$^{-3}$ in 2017 and 2018 (Fig. 4B). *Bythotrephes* production varied through the time series, ranging from 2.7 to 31 µg m$^{-3}$ dd$^{-1}$ (Fig. 4D). The first instance of *Bythotrephes* detection in our annual zooplankton tows ranged from day-of-year 121 (May 1, 2015) to day-of-year 173 (June 21, 2012), during the early season clear water phase (Fig. 3). *Bythotrephes* is a large and fast-growing zooplankton that preys voraciously on other zooplankton[44,45]. It can reduce native crustacean zooplankton community biomass by 40–60% and production by 67%[17]. Additionally, *Bythotrephes* can consume >100% of native cladoceran production in a given time interval[28,31]. In Lake Mille Lacs, the native cladoceran zooplankton density and production decreased rapidly (by 93% and 87%, respectively, relative to pre-*Bythotrephes* levels in 2006–2010) after 2010 (Fig. 4C, E). Native copepod zooplankton density also decreased

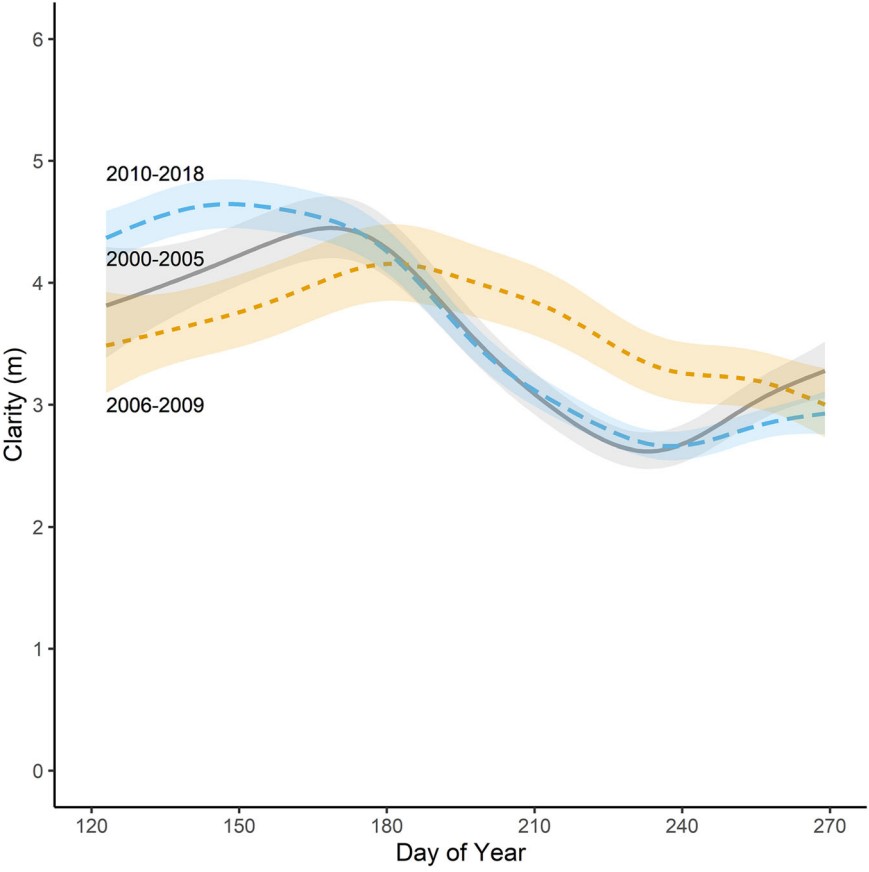

**Fig. 3 Water clarity (m) patterns in Lake Mille Lacs prior to *Dreissena* or *Bythotrephes* invasion (2000–2005), post-*Dreissena* invasion (2006–2009), and post-*Dreissena* and *Bythotrephes* invasion (2010–2018). Estimates are simulated from the generalized additive model.** Confidence ribbons are ±1 standard error.

but did not do so until after 2011, one year after the observed reduction in cladoceran zooplankton (Fig. 4C). Relationships between native zooplankton groups and *Bythotrephes* suggest strong non-linear top-down effects (Fig. 5), similar to these same relationships in Lake Michigan[26]. We divided the copepods into two groups, cyclopoids (mainly omnivores) and calanoids (mainly herbivores) for production estimates to avoid double counting some of the productivity through predation. Cyclopoid production steadily decreased after 2010 (Fig. 4E), while calanoid production increased briefly in 2010–2011 before subsequently decreasing. Coincident with declines in native zooplankton density and production, their predicted filtering rates plummeted (Fig. 4G). While in other systems invaded by *Bythotrephes* the native zooplankton migrated vertically to cooler strata, reducing their production[46], this was unlikely here because Lake Mille Lacs does not thermally stratify (Fig. 2)[20,21].

**Water column filtering rates**. Given the near ubiquitous increase in water clarity in *Dreissena* infested waters, it is surprising that summer water clarity remained stable after the establishment of *Dreissena* in Lake Mille Lacs (Figs. 3, 4F). The co-invasion of *Bythotrephes* and *Dreissena*, and their synchronous but opposite effects on food-web processes, is the most feasible explanation for this observation. To test this hypothesis, we compared modeled filtering rates of the native zooplankton versus *Dreissena* (Fig. 4G). Despite significant changes in densities of *Dreissena*, *Bythotrephes*, native cladocerans, and copepods, we did not find a statistically measurable net change in water column filtering by phytoplankton grazers, chlorophyll *a*, or water clarity in the lake (Fig. 6). We suggest that this was due to the shrinking top-down

grazing pressure by the native crustacean zooplankton being equally compensated for by expanding top-down grazing pressure by *Dreissena* during the half decade following the co-invasion[37]. Beginning in 2011, *Dreissena* filtering rate surpassed native zooplankton filtering rate and over the next several years progressively filled this functional niche at the same capacity formerly occupied by the native crustacean zooplankton community (Fig. 4G). Our analysis shows that community filtering rates have remained stable through time but have been reallocated from pelagic to benthic taxa.

**Bottom-up and top-down mechanisms**. We tested the two alternative hypotheses that *Dreissena*-driven changes to water clarity were buffered by bottom-up (phytoplankton growth) or top-down (fish predation) processes resulting in no net change in clarity. There is no evidence that phytoplankton blooms are driving the lack of change of water clarity in Lake Mille Lacs. While phytoplankton blooms were captured during the monitoring period, as indicated by large chlorophyll *a* values (variability in Fig. 4F), they were mostly singular annual events between May and September, with the exception of 2017, having two blooms. Although it is possible that primary production could increase without an increase in chlorophyll *a* concentrations, there was no trend in total phosphorus concentrations that would drive changes in primary production (Fig. 4H, Mann Kendall trend test, $p = 0.430$). It is possible that these datasets did not capture increases in nutrient loading to the lake that would offset increases in water clarity driven by *Dreissena* because of the frequency, timing, or location of monitoring. In Minnesota lakes, significant increases in phosphorus concentrations are associated with increases to agriculture

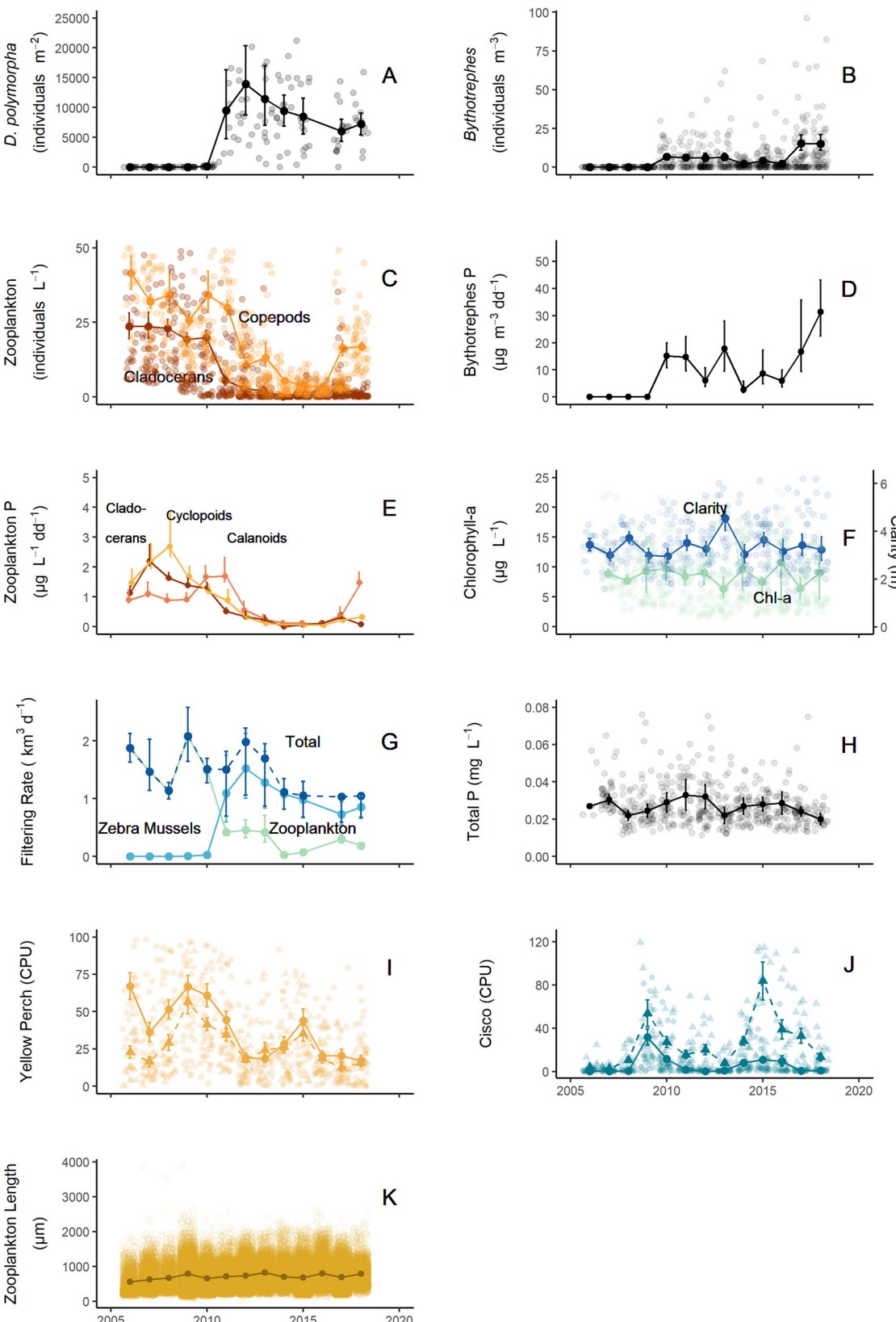

or developed land cover to 40% or greater of total watershed area[47]. In the Lake Mille Lacs watershed, there was not a significant change in land use from 2001 to 2016, with anthropogenic disturbance (sum of agriculture and developed land covers) decreasing from 10.0% to 8.9% of the watershed area during that timeframe (Supplementary Fig. 3). These values are much lower than the 40% threshold, suggesting that phosphorus load related to landuse in the watershed did not change during this study. Additionally, we found no difference in cumulative growing degree days among the different invasion status groups that would drive increases in productivity (Mood's median test, chi-squared = 0.689, $p = 0.837$, Supplementary Fig. 4).

**Fig. 4 Characteristics of Lake Mille Lacs, Minnesota, 2006–2018. A** Median *Dreissena* density (individuals m$^{-2}$, $n = 169$). **B** Median *Bythotrephes* density (individuals m$^{-3}$, $n = 708$). **C** Median native zooplankton density (individuals L$^{-1}$; cladocerans, brown; copepods, orange; $n = 658$). **D** Median *Bythotrephes* production from May-September (µg m$^{-3}$ degree d$^{-1}$). **E** Median native zooplankton production from May-September (µg L$^{-1}$ degree d$^{-1}$; cladocerans, brown; calanoids, dark orange; cyclopoids, light orange). **F** Median June-August chlorophyll *a* concentration (µg L$^{-1}$, green, $n = 502$) and median May-September water clarity (m, blue, $n = 346$). **G** Median filtering rate (km$^3$ d$^{-1}$; *Dreissena*, light blue; zooplankton, teal; summed, dark blue). Error bars represent the 95% C.I. in (**A–G**). **H** Median total phosphorus (mg L$^{-1}$, $n = 621$). **I** Mean yellow perch catch per unit effort (CPU) from nearshore (solid circle, $n = 416$) and offshore (dashed triangle, $n = 260$) gillnets. **J** Mean cisco catch per unit effort (CPU) from nearshore (solid circle, $n = 416$) and offshore (dashed triangle, $n = 260$) gillnets. **K** Mean body length (µm) of native zooplankton, $n = 89,810$. Error bars represent 1 standard error in (**H–K**).

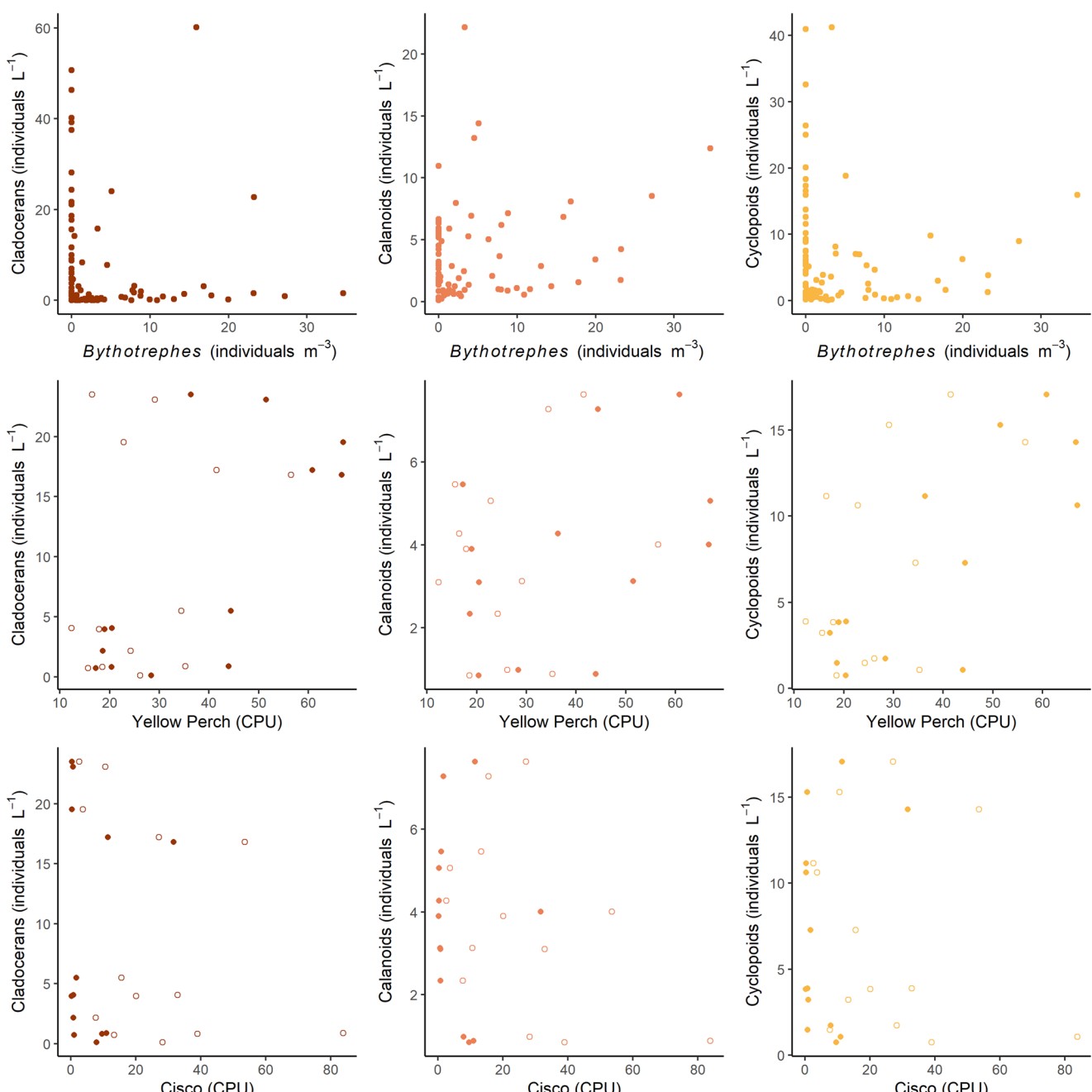

**Fig. 5 Relationships (2006–2018) between abundances of native zooplankton groups (cladocerans, calanoids, and cyclopoids, individuals L$^{-1}$) and predators of zooplankton (*Bythotrephes*, $n = 87$, individuals m$^{-3}$; yellow perch and cisco, $n = 13$ observation for each fish-location combination, CPU, catch per unit effort).** Open symbols represent fish data from offshore gill nets, and closed symbols represent fish data from nearshore gill nets. The linear relationships between nearshore yellow perch and both cladocerans and cyclopoids were significant ($p = 0.0053$ and $0.0011$, $r = 0.722$ and $0.799$, respectively).

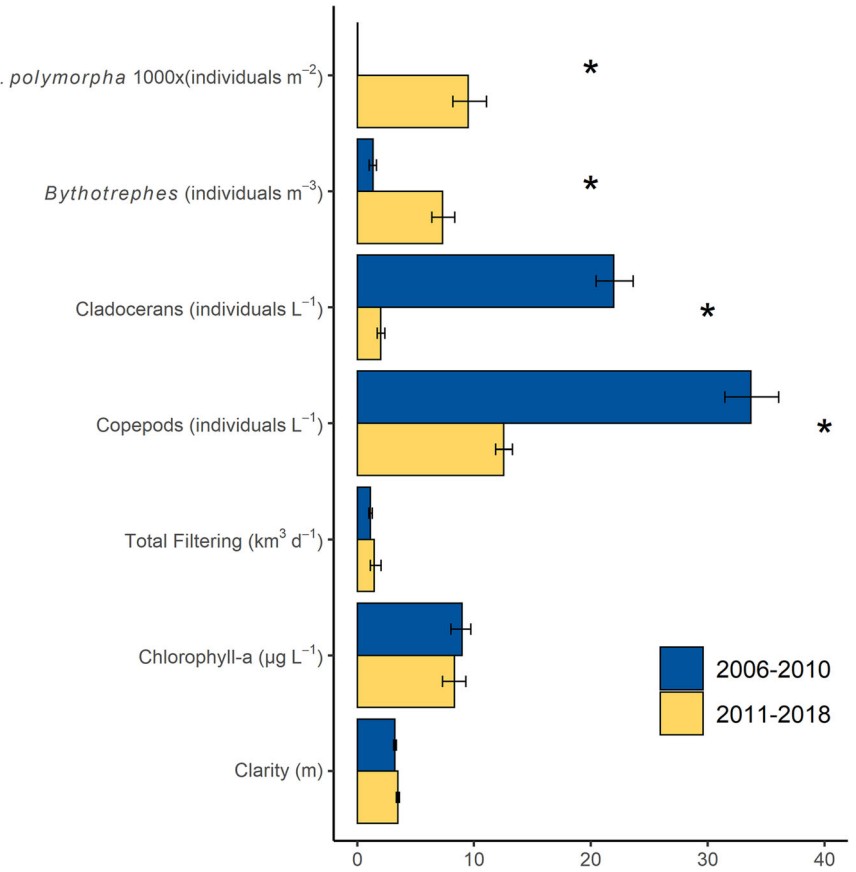

**Fig. 6 Comparisons of community attributes and water clarity in Lake Mille Lacs related to invasion.** Densities of two invasive invertebrates (*Dreissena*, 1000 * individuals m$^{-2}$, and *Bythotrephes*, individuals m$^{-3}$) before (2006–2010) and after (2011–2018) both populations expanded. The densities of native zooplankton (cladocerans and copepods, excluding *Bythotrephes* individuals L$^{-1}$) decreased in response to co-invasion, but total water column filtering (sum of herbivorous zooplankton and *Dreissena* filtering, km$^3$ d$^{-1}$) did not change. As a result, there was no change in median chlorophyll *a* (μg L$^{-1}$) or median water clarity (m) despite large food web perturbations. Bars are medians of bootstrapped estimates. Error bars represent the 95% C.I., and asterisks represent significant changes ($p < 0.05$) to a variable.

In contrast to a bottom-up effect buffering changes in water clarity, we considered the possibility of top-down fish impacts to clarity. If planktivorous fish increase, we expect that native zooplankton would be subject to increased mortality, as they are in the presence of *Bythotrephes*, releasing phytoplankton from grazing pressure. Yellow perch (*Perca flavescens*) and cisco (*Coregonus artedi*) are the largest (body length) planktivorous fish in Lake Mille Lacs. We found a trend of decreasing yellow perch (Fig. 4I, Mann Kendall trend test, $p = 0.009$) catch rates, and no trend in the catch rates of cisco (Fig. 4J) during the time series. We did not find evidence of top-down control by either planktivorous fish species on native zooplankton groups (Fig. 5). There was evidence of bottom-up control of yellow perch in nearshore gill nets by both native cladocerans and cyclopoids. Additionally, increased fish predation would shift the zooplankton size structure towards smaller individuals[48,49]. We found that mean native zooplankton body size increased over the time series (Fig. 4K, Mann Kendall trend test, $p = 0.044$), consistent with the response of native zooplankton to *Bythotrephes* predation[50,51]. Although mean body size increases after *Bythotrephes* establishment in the lake, it is variable. It is possible that a release from fish predation (Fig. 4I) is also driving the size structure of the native zooplankton community[52].

**System productivity.** While there was no change in summer water clarity or water column chlorophyll *a* in Lake Mille Lacs over the 19-yr time series (Fig. 3), the lake still followed the predicted trajectory of benthification typical of *Dreissena* infested

waters[22,24,25]. Specifically, pelagic biomass and production of zooplankton declined commensurate with proliferation of benthic biomass and production of *Dreissena*, shifting the process of conversion of phytoplankton to animal biomass from the offshore pelagia to the nearshore benthos (Fig. 4A, C)[22]. This redistribution of energy and nutrient flow is predicted to impact higher trophic levels[53,54], and in Lake Mille Lacs young-of-year walleye (*Sander vitreus*) are currently growing slower than they were historically, following patterns observed in other *Dreissena*- and *Bythotrephes*-infested Minnesota lakes[55].

**Combined effects of invasive species.** Our data suggest that the impacts of *Dreissena* and *Bythotrephes* on water clarity were symmetrically antagonistic, cancelling one another at the scale of the whole ecosystem. While we predicted antagonism based on first principles of food-web networks, our results offer important confirmation of the expected response of a native plankton community to these two invaders. In addition to the antagonistic effects on water clarity caused by *Dreissena* filtering and *Bythotrephes* predation, the dyad appears to have exerted additive (or partially additive) impacts on the diversity and abundance of native zooplankton (Fig. 4C, E). *Bythotrephes* characteristically suppress cladocerans and cyclopoid copepods through predation and competition. However, some cladocerans (most notably *Daphnia mendotae* and *Holopedium*) and calanoid copepods coexist with *Bythotrephes* and may increase in density post-invasion as a response to reduced competition

for phytoplankton[17,56]. In Lake Mille Lacs, the comprehensive collapse of 16 of 20 native zooplankton taxa (including *D. mendotae* and *Holopedium*) is strong evidence that *Bythotrephes* was not the sole agent of change. The brief, 2-year surge in calanoid copepods in 2010–2011 may reflect compensatory reductions in competition for phytoplankton but their subsequent collapse (Fig. 4C, E), despite no change in phytoplankton, demands an alternative mechanism. We suggest that continued suppression of calanoid copepods, particularly nauplii, may be at the hands of *Dreissena* predation[24] and chemicals (kairomones)[57] exuded by *Bythotrephes*, and that the cladocerans *D. mendotae* and *Holopedium* may suffer from increased competition among fish and invertebrate predators for the remaining zooplankton resource.

It is feasible that an asynchronous invasion scenario, where one or the other nonindigenous species gains establishment first, could bring different outcomes depending on the strength and direction of priority effects[9,14,58,59]. Asynchronous invasion where *Dreissena* arrives first could affect establishment of *Bythotrephes* through benthification of nutrients that inhibits cladoceran zooplankton growth, and enhanced water clarity that allows visual planktivorous fish to better patrol for large-bodied zooplankton. Alternatively, asynchronous invasion where *Bythotrephes* arrives first could reduce crustacean zooplankton density, enhance phytoplankton abundance, and enhance the food base for *Dreissena* improving their growth potential.

We recognize that long-term ecological research, like ours, often relies on BACI (Before-After-Control-Impact) study designs that sacrifice ecosystem replication for continuity and comprehensiveness. In our circumstance, ecosystems with *Dreissena*-only invasions serve as a form of "control"[60], and as we highlighted above, water clarity trends in Lake Mille Lacs depart measurably from most such 'control' ecosystems after *Dreissena* invasion.

**Management implications**. The uncertainty of predicting the outcomes of multiple invaders, both synchronous and serial, is without question a major challenge that researchers and managers face[61,62]. In Lake Mille Lacs, we did not observe the expected clearing of the water column due to *Dreissena*, but the translocation of resources and energy pathways expected in response to both invaders did occur. For natural resource managers this is especially important as they are tasked with the challenge of managing various species within these changing communities and managing the expectations of regional stakeholders regarding ecosystem services. These findings emphasize the importance of long-term ecological research, work often pursued by agency scientists and/or supported with public funds, to inform adaptive management plans[63–65].

Shifts in water clarity and phytoplankton biomass have long been heralded as reliable signs of ecosystem change in lakes and streams. Here we provide an example of alterations in the composition of consumer species and pathways of energy flow in a large-lake food web that triggered no significant shifts in water clarity and phytoplankton biomass. Rather, the dual timing of the co-invasion by *Dreissena* and *Bythotrephes*, and the compensatory nature of their impacts, left the food web of Lake Mille Lacs transformed under a veil of relative stability in water clarity and phytoplankton biomass. These results demonstrate the vulnerability of using only simple metrics of health (e.g., water clarity) as harbingers of ecosystem change in response to stressors, in this case invasive species.

## Methods

**Field observations**. Lake Mille Lacs is a large, shallow lake (surface area = 520 km$^2$, mean depth = 8.7 m, maximum depth = 12.8 m) in central Minnesota, USA.

The lake is mesotrophic, and water clarity increased from the mid-1970s to the mid-1990s, possibly as a result of the 1970 Clean Water Act. The lake is a culturally and economically important walleye (*Sander vitreus*) fishery that is managed cooperatively by eight Ojibwe tribes and the Minnesota Department of Natural Resources (MNDNR). Besides *Dreissena* and *Bythotrephes*, other invasive species in Lake Mille Lacs include common carp (*Cyprinus carpio*) and Eurasian watermilfoil (*Myriophyllum spicatum*).

*Dreissena* population estimates were determined annually after their first detection in 2005 using a variety of methods. In 2005, divers visually assessed ~60 areas in the lake for about 50 min at each location. In 2006, 20 monitoring stations were established and all *Dreissena* within ~60 cm of a 183-m transect were enumerated by two divers swimming the length of the transect. The number of monitoring stations ranged from 9–22 from 2007–2013. By 2011, *Dreissena* densities were too high to count while diving, so samples were collected and counted at the surface[37].

The MNDNR has a long-term monitoring program for Lake Mille Lacs, and after *Dreissena* was first detected in 2005 the agency added monthly zooplankton monitoring in 2006. Staff collected standard zooplankton tows of the whole water column using a zooplankton net with 80-µm mesh and a 13-cm diameter opening in 2006–2008, and a net with 80-µm mesh and a 30-cm diameter opening in 2009-2018. Staff collected zooplankton once monthly from May-September from 2006 to 2009 at nine stations (Fig. 1). In 2010, MNDNR staff began collecting zooplankton semi-monthly in May and June to better capture the rapidly changing demographics of the spring zooplankton community. Zooplankton were preserved in 80% ethanol until they were processed in the laboratory[66,67]. The same researcher (J. Hirsch) conducted all measurements throughout the study. Annually between 2060 and 12,600 individual zooplankton were identified, enumerated, and measured to the nearest 0.1 mm, allowing us to reconstruct the demographics of the populations and use data-intensive calculations of their energy flux though time (details below). We calculated annual estimates of median zooplankton density by resampling the monthly density estimates using bootstrapping (1000 iterations)[68] and reporting the average of the monthly bootstrapped estimates. We estimated the 95% confidence interval (C.I.) using the percentile method (2.5th and 97.5th percentiles, throughout)[68]. Divers examined the substrata underneath several of the zooplankton stations. The underlying benthos of one station was colonized by *Dreissena*, five stations were not colonized, and the colonization statuses of the other three stations are unknown.

In 2006–2018, MNDNR staff collected water samples from the top 2 m of the lake surface using a polyvinyl chloride tube for chlorophyll *a* and total phosphorus (TP) analyses at the nine stations during zooplankton sampling (Fig. 1). Prior to 2006, water chemistry samples were collected at 1–5 stations around the lake, typically once during the summer. Lake Mille Lacs is well mixed and rarely stratifies in the summer (Fig. 2)[20,21], so surface water chlorophyll *a* and TP concentrations represent water column conditions. Water samples were stored on ice in the dark and shipped overnight to the Minnesota Department of Agriculture Laboratory for chlorophyll *a* analyses using spectrophotometry[69] and TP analyses by colorimetry[70]. Median open water chlorophyll *a* and TP estimates were calculated using daily median values to calculate the monthly median before summarizing for each year to dampen effects of uneven sampling due to increasing May and June sampling to twice a month (2010–2018). The TP dataset included samples that were below the detection limit (0.010 mg L$^{-1}$), so we used the median, instead of the mean, to describe the center of data[71]. We estimated daily water clarity using Secchi disc depth (0.1 m) collected from May-September from multiple sampling programs and accessed through the Minnesota Pollution Control Agency (MPCA) data portal (www.mpca.state.mn)[72], as well as data collected by MNDNR biologists alongside zooplankton tows. We estimated median annual chlorophyll *a* concentrations and clarity measurements, and we computed 95% C.I. by resampling the average of daily observations for each year (1,000 iterations) using bootstrapping[68].

MNDNR staff collected continuous water temperature data at 2-m depth from 2001-2018, excluding 2008 (HOBO temperature logger, Onset, Bourne, MA). We imputed missing data by assessing the linear relationship between our measured mean daily water temperature data and modelled mean daily water temperature data at 2-m depth[73] ($r = 0.995$, $p < 0.001$) and applying a correction to the modelled data (T = T$_{modelled}$ −0.71) before including it in our dataset. We calculated cumulative degree days (base 0 °C) for each year in the study (Supplementary Fig. 4). We used two-sided Mood's median test with Monte Carlo simulation[74] to compare cumulative growing degree day (base 0 °C) at day-of-year 270, the end of our open water time series each year, among the three different invasion statuses.

Discrete sampling events may not capture changes in nutrient inputs to a lake, so we also examined land cover changes in the watershed of Lake Mille Lacs from 2001-2016 using the National Land Cover Dataset[75] to assess if we were missing any watershed-scale changes that might impact nutrient inputs to the lake (Supplementary Fig. 3). We compared land cover in the watershed to a published relationship between land cover and total phosphorus levels in a study of over 1300 natural lakes in Minnesota[47].

Fish data were collected annually in late September by MNDNR Fisheries staff using guidelines of professional fisheries societies[76]. Fifty-two nets were set at established locations throughout the lake and left overnight, with set times of approximately 20 h. Each net consisted of five 15.2-m long, 3.1-m deep panels of

multifilament nylon, with mesh sizes of 1.91-, 2.54-, 3.18-, 3.81-, and 5.08-cm that were effective at catching individuals >150 mm length. Fish were returned to the laboratory where they were counted, measured (mm), and weighed (g). We reported mean annual CPU (catch per unit effort) and standard error. We used abundance relationships between zooplankton predators (*Bythotrephes* and fishes) and three zooplankton functional groups to examine top-down versus bottom-up impacts between trophic levels. The plots of *Bythotrephes* versus zooplankton are the means of all stations by sampling date, while the plots of fish versus zooplankton are annual means.

**Statistics and reproducibility**. We used a hierarchical generalized additive model (HGAM)[77] to model the clarity data and test for the effects of invasion status (neither invasive species, *Dreissena* only, or *Bythotrephes* and *Dreissena*) on clarity. We assigned the three levels of the "invasion status" variable based on the timing of the first known observation of *Dreissena* (assigned to 2006, although four individuals were found in 2005) and *Bythotrephes* (assigned to 2010, although 2 individuals were found in September 2009) in the lake. We accounted for seasonal patterns in clarity by including a 'day-of-year' variable and annual variability by including a random 'year' variable, both with a global smoother. We accounted for variability among stations by including a 'station' variable with a global smoother, although we are not sure if this variable represents spatial variability or variability associated with data collected by the public via a community science program. We also included a 'day-of-year:invasion status' interaction, which applies different smoothers for each level of the 'invasion status' variable. We fit the HGAM using an inverse Gaussian distribution and a log-link function. We used simulated data from the final GAM to assess spatial patterns in the clarity measurements. We held covariates constant and fit the model for individual sampling stations. We plotted the resultant clarity measurements against multiple metrics (distance from nearest known zebra mussel bed, distance from shore, and water depth)[78,79] calculated in ArcGIS Desktop 10.6[80] (Supplementary Fig. 2). We used the R package 'mgcv' version 1.8-36 for HGAM analyses[81].

We calculated secondary production of the native zooplankton populations using the size-frequency method[82] and summed individual population estimates into higher levels of taxonomic organization. We estimated median annual production and uncertainty (95% C.I. from percentiles) using bootstrapping (1000 iterations)[68] of the zooplankton population densities in 100- or 200-μm interval size bins on each sampling date (code modified from the literature[83]). We applied a size- and temperature-dependent correction for development time using temperature data described above[84]. We standardized secondary production by degree-day using water temperature data (base 0 °C) to account for temperature variability among years[85]. We calculated the rates of water column filtering for native zooplankton[86] and *Dreissena*[42] to compare their effective removal of phytoplankton from the water column. We estimated the error (95% C.I. using percentiles) associated with the filtering rates by bootstrapping (1000 iterations)[68] the zooplankton biomass and *Dreissena* densities to use in the calculations of the filtering rates. Because Lake Mille Lacs is polymictic, we applied the formula for mixed lakes for the *Dreissena* filtering rates[42]. To compare the effects of invasion to conditions in the lake prior to invasion, we grouped the data for years into early (2006–2010, *n* = 5 yr) and well established (2011–2018, *n* = 8 yr) categories, based on the timing of expansion of both *Dreissena* and *Bythotrephes* populations (2011 was the first year that both population densities reached high levels). We compared parameters between these two groups by averaging the annual bootstrapped (1000 iterations) values for each group and reported the median and 95% C.I. of those averaged estimates[68]. We considered differences between the groups as significant (α = 0.05) if the 95% C.I. did not overlap. For all parameters (*n* = 13 yr) the data were resampled where noted in the methods. To assess the individual parameters (TP concentrations, fish catches, and mean native zooplankton length) for monotonic trends through time, we aggregated the data at the smallest possible time interval (usually day) and assessed trends using a one-sided Mann Kendall test[71]. We used Mood's median test to compare cumulative annual temperature (base 0 °C) among invasion statuses on day-of-year 270. We produced all figures and conducted all analyses in R[87].

**Reporting summary**. Further information on research design is available in the Nature Portfolio Reporting Summary linked to this article.

## Data availability
Fish, lake water chemistry, zebra mussel, zooplankton, and water temperature data are available from the corresponding author on reasonable request. Water clarity data are available in the Minnesota Pollution Control Agency Surface Water Data repository with the unique waterbody identifier "48-0002-00". Landuse data are available from the National Land Cover Database (www.usgs.gov). Minnesota-specific spatial data used for mapping are available from the Minnesota Geospatial Commons (www.gisdata.mn.gov). Modelled water temperature data, which we used to impute missing values from water temperature data collected by sensors, are available from the United States Geological Survey ScienceBase Catalog (www.sciencebase.gov), and we used spatial data from that same source for the map of North America. Figures 1, 2, 4, 5, 6, and Supplementary

Figs. 2–4 use data from the sources listed above. Data for sampling station locations (Fig. 1), Figs. 2–6, and Supplementary Figs. 1–4 are available in the Supplementary Data.

## Code availability
Code for Figs. 2–5 and Supplementary Figs. 1–4 are included as Supplementary Note. Other computer codes used for this work are available from the corresponding author upon request.

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

## Acknowledgements

This project was funded by the Minnesota Department of Natural Resources. We thank all of the MNDNR staff who collected data that we used in this study. E. Jensen and R. Rezanka helped with the zebra mussel diving surveys. J. Walsh provided input on statistical modeling. A. Williquett assisted with spatial analyses. T. Ahrenstorff, J. Hoxmeier, J. Reed, P. Schmalz, and M. Treml provided constructive feedback. We are grateful to three anonymous reviewers for thoughtful comments that improved this manuscript.

## Author contributions

D.B. and H.R. conceived the study; J.H. produced the zooplankton data; T.J. and G.M. produced the zebra mussel data; T.J. produced the fish data; H.R. analyzed the data, modeled zooplankton production and invertebrate filtering rates, and produced the figures; H.R. and D.B. designed the figures; H.R. and D.B. drafted the manuscript, with contributions from T.J., G.M., and J.H. T.J. provided the photograph.

## Competing interests

The authors declare no competing interests.
