## [Peer Review File · Communications Biology]

Reviewers' comments:

Reviewer #1 (Remarks to the Author):

General comments:

The authors reported very interesting findings pertaining to the response of water clarity and primary production to co-invasions of zebra mussels and water flea. I enjoy reading this work, and I confirm its significance in advancing our understandings of the impacts of invasive species on ecosystems from a broad perspective. Overall, the methods are sound though I think there is room to improve the data analysis. However, I have several concerns, and some are critical after I read this manuscript. I believe this work can be improved by addressing the following issues:

1) A critical prerequisite for the compensatory influence of the water flea on the mussel clearance is that the lake is well mixed during in May-Sep; however, the authors did not provide data to support this. Given that only the upper layer of the water column was sampled for Chla, it is very critical to ensure that the water column is well mixed.

2) Because the influence of zebra mussel might be constrained to certain areas in large lakes, some decoupled sampling points between zebra mussel and Chla/Zooplankton might harm the power of the statistical analysis, especially those from the central lake (presumably deeper). E.g., thinking about points at which no mussels but water flea existed, the latter would potentially increase the Chla. Including these points in the analysis might compromise the effect of mussels from nearshore sites as they do reduce the Chla and increase water clarity; It would be worth trying a sub dataset to investigate this;

3) I believe that more specific analysis is needed to better dig the time series data, especially the water clarity, as it seems that the water clarity did increase following the invasions. A time-series linear model may help uncover this trend;

4) The authors should dig deeper into their results. For example, the water flea density was not very high until 2017-18, and what caused this and the potential impacts on the compensatory relationship between zebra mussel and water flea should be discussed;

5) Some interpretations seemed weak due to a lack of statistical results, and I suggest linking discussion more tightly with results.

Specific comments

L44: The impact of Dreissena increases rather than reduces the water clarity.

L55: It would be very helpful if the mixed effects on phytoplankton being briefly summarized here.

L98: Any evidence that the lake was not stratified?

L137-139: I think this interpretation is very weak. If we focus on the period of 2008-2018, during which the yellow perch declined, it seems that the body size of zooplankton did not change over time. The density of water flea was medium to low during 2010-2016, during which the yellow perch and cisco were decline as well. Why?

Providing depth contour of the lake will be very helpful.

According to Fig. 1, the surveys of mussels were largely restrained to nearshore – possible shallow region – while the zooplankton and Chla survey spread out across the lake. This raises a question: the decoupling of the zebra mussel and zooplankton might underestimate the impacts of zebra mussels on Chla or overestimate the compensatory effect of the water flea on Chla. The authors should justify their selection or re-calculate the effect of mussels by focusing on the coupled sites (i.e., zebra mussel vs. Chla) only.

I suggest the authors combining the time series of some subpanels in Fig. 2 and Fig. 4 into a single panel for better comparisons among tropic levels.

Reviewer #2 (Remarks to the Author):

Review of COMMSBIO-21-0240-T

Brief summary

The manuscript reports on a study that takes advantage of the co-invasion of a lake in Minnesota by two aquatic invaders– the spiny water flea (*Bythotrephes longimanus*) and the zebra mussel (*Dreissena polymorpha*) – to examine the co-effects of multiple invasive species on a native ecosystem. This was done using an annual, long-term observational monitoring program of the lake's food web. The manuscript reports that water clarity in the lake remained stable after the establishment of *Bythotrephes* and *Dreissena*, which is surprising given that *Dreissena*-invaded waters typically increase in water clarity due to *Dreissena*'s filtering activities. The authors conclude that the maintenance of water clarity can best be explained by synchronous but opposite effects of *Bythotrephes* and *Dreissena* on food web processes, which they test by comparing modeled filtering rates of the native zooplankton versus *Dreissena*.

Overall impression

Broadly and conceptually, this is an interesting case study on the effects of multiple invasive species on ecosystem processes. More specifically, the reported findings could have important applications as the two invasive species in the study are two of north America's most widespread and problematic. The long-term dataset (13 years) is a strength as there are several years sampled before and after the invasion events. Finally, the manuscript is generally well-written and well-referenced. However, like all 'natural experiments' the study does have some limitations. Further, the background information provided is quite dismissive of important foundational and ongoing work on multi-species invasions and their effects on ecosystems. The manuscript could be strengthened by (1) couching the work in a more balanced and accurate framework of multi-species invasions, and (2) better acknowledging the limitations of unreplicated or uncontrolled 'before-after' natural experiments.

Specific comments

(1) There has been much conceptual, methodological, and empirical work done on multi-species invasions – dating back to Simberloff and Von Holle's classic 'Invasional meltdown' paper in *Biological Invasions* in 1999. Therefore, I found lines 28-32 on Page 2 a bit dismissive of this body of work, and a somewhat inaccurate characterization of the field. There has even been a meta-analysis on multi-species invasions in the last five years (Jackson 2015 *Ecology*). See also the experimental test and conceptual framework by Braga et al. 2019 *Hydrobiologia*; the methodological and experimental work by Pearson et al. 2016 *Ecological Applications*; and empirical work on multi-species invasions by e.g., Adams et al. 2003 *Ecology Letters* and Montgomery et al. 2012 *Biological Invasions*.

(2) Whereas long-term monitoring datasets enabling us to detect anthropogenic perturbations and their effects on natural systems are critical and all-too-rare, they are limited in their own right. I applaud the monitoring work done here and appreciate the amount of work that went into this study and think we gain some real insight from it. However, I do think the manuscript could be strengthened by acknowledging that some caution should be taken when using, or interpreting results from, a BACI-type design when no control system was available (i.e., Before-After only as you have here). See e.g., critical work by Underwood on this topic: *Ecological Applications* 1994, *Marine and Freshwater Research* 1991, etc., as well as that by Benedetti-Cecchi 2001 *Ecological Applications*, and Stewart-Oaten et al. 1986 *Ecology*)

Reviewer #3 (Remarks to the Author):

The authors provide a relatively high-level analysis of potentially competing effects of two invasive species on water quality and lower trophic level indicators in Lake Mille Lacs, MN. The authors purport to demonstrate a rare stability of phytoplankton biomass and water clarity despite a concurrent invasion of zebra mussels and the spiny water flea. The lack of change in chlorophyll a and water clarity is attributed to grazing control of phytoplankton by zebra mussels and a concurrent relaxation

of phytoplankton due to predation on herbivorous zooplankton by the spiny water flea. While this is a reasonably novel construct from first principles and would be of broad relevance to invasion ecology and lake management, I do not find the evidence overly compelling.

First, I find the sole focus on the annual timescale obscures potential information that may well (or not) support the authors conclusions. For example, at a sub-annual scale, do patterns of chl_a/clarity covary with anticipated time periods of 1) production and settling of dreissenid veligers or 2) peak abundance of Bythotrephes (and presumed predation of herbivores)? Comparisons of annual means subsumes variability that may well be instructive with respect to the claims made here.

Second, the authors present a well supported argument that Bythotrephes has likely relaxed some grazing pressure on phytoplankton as all reported zooplankton metrics are consistent with known impacts of Bythotrephes invasion. However, they also argue that they can rule out changes in phytoplankton growth yet do not present information on primary production rates or other proxies (13C data? C:Chl ratios?) needed to evaluate it. The presence of periodic blooms does little to inform on growth rates of edible phytoplankton species. A passing reference is made to possible changes in non-point source nutrient inputs as a potential buffer to mussel grazing, but only present point in time measures of chl_a as evidence this is not occurring. Such reasoning ignores hydraulic forcings, circulation and wind and waves as possible reasons why one might also find elevated chl_a in some places. The authors also argue that water chemistry data collected at an outflow site that only partially overlaps the study period is somehow representative of nutrient inputs to lake infested with zebra mussels, which is a significant stretch and at odds with the general body of evidence relating mussel impacts on nutrient stores in lakes. This part of the hypothesis unfortunately remains speculative in my opinion without more robust data.

Lastly, some of this data appears in a recently published paper (Jones and Montz 2020; Biol. Inv. 9(4):772-792) where those authors have come to similar conclusions and raised the idea that Bythotrephes may be responsible for the apparent lack of change in clarity and chl_a. I would recommend the authors acknowledge that work should they decide to revise the paper.

Specific comments

Statistical methods – While I can understand the bootstrapping approach used here, I wondered if a mixed effect modeling framework might provide additional insight into the sequence of changes and potential confounders. It can easily be done from a pre/post perspective or change over time. Further, I wonder if bootstrapping across data derived from one source (zooplankton) vs others which appear to come from multiple sources (i.e. secchi disc) is appropriate. A mixed model framework can handle that aspect fairly easily, while still providing estimates of the main effects. It appears Jones and Montz (2020) Biol. Inv. 9(4):772-792 have already taken this approach for some of this data with respect to mussel effects on water clarity and chl_a.

Line 33 – can you confirm only *D. polymorpha* present (i.e. no *D. bugensis*)?

Line 71 – typical time to peak population of *D. polymorpha* in NA is ~ 2.5 yrs – why the long lag here? Possible that population undersampled?

Line 113 – ruling out bottom – up processes would probably entail either modeled primary production or some measure of primary production no? blooms (depending on species) may not necessarily represent edible algae, and they tend to represent a small time frame of a typical season.

Lines 121-125 – why use chl_a at point sources to infer nutrient loading has not increased? Would not actual load estimates be a better proxy? Furthermore, one assumes that water still circulates within the lake so point in time chl_a samples really don't provide convincing evidence one way or the other.

Lines 141-142 – are there other data to support or refute the clarity hypotheses? i.e. no change in macrophyte cover? Extent? Depth of colonization? Simon and Perleberg (2015) reported no change in maximum roo

Lines 125-128 - The time frame of water quality at an outlet site between 1955 – 2010 tells me little about trends in nutrients entering the study lake from 2006-2018. A decline in outflow concentrations is not unexpected post colonization by mussels.

Line 329 – why not include GDD and solar insolation as other proxies of energy flux – such variables are known to produce coherent patterns in biological response variables (i.e. chl_a) – could these also

explain why the signal in chl_a is weak?

Line 336 -338 – Chl_a can still vary vertically even in well mixed systems, particularly during quiescent periods. Do you have additional information to support this? Perhaps fluorometer data? Chl_a data at different depths? Further; were samples extracted using the same solvent across the time series?

Lines 347-349 - The exclusion of nearshore chl_a collected near point sources is somewhat confusing – I get the desire to remove confounders, but it is not like dreissenids eschew a free meal – perhaps a note to say that grazing rates could be underestimated by doing this?

Fig 2. It appears as if there is a slight upward trend in clarity but nothing obvious in chl_a. If grazing is indeed best placed to explain the lack of pattern, could a shift to smaller cells account for a small increase in clarity while keeping chl_a the same?

Response to Reviewers

Simultaneous invasion decouples zebra mussels and water clarity

Rantala et al.

COMMSBIO-21-0240-T

Reviewer #1:

1) A critical prerequisite for the compensatory influence of the water flea on the mussel clearance is that the lake is well mixed during in May-Sep; however, the authors did not provide data to support this. Given that only the upper layer of the water column was sampled for Chla, it is very critical to ensure that the water column is well mixed.

We addressed this issue two ways. We included heat maps for the lake for all years where we had sufficient temperature profile data (Figure 2). We also cited work by Fang et al. (2015) and Jacobson et al. (2008) that calculated the geometry ratio of Lake Mille Lacs and relates that metric to stratification stability and duration (Stefan et al. 1996) in Lines 48-50.

2) Because the influence of zebra mussel might be constrained to certain areas in large lakes, some decoupled sampling points between zebra mussel and Chla/Zooplankton might harm the power of the statistical analysis, especially those from the central lake (presumably deeper). E.g., thinking about points at which no mussels but water flea existed, the latter would potentially increase the Chla. Including these points in the analysis might compromise the effect of mussels from nearshore sites as they do reduce the Chla and increase water clarity; It would be worth trying a sub dataset to investigate this;

The zooplankton/water quality stations in the lake are pelagic, and only 1 of 9 is known to be in a zebra mussel-rich area. Five of the stations do not have zebra mussels at them (via. diving surveys), and the presence/absence of zebra mussels at the remaining three stations is unknown. We explored some additional statistical models to evaluate spatial patterns, and there was a spatial effect among individual stations but there was not a difference in stations with or without zebra mussels. We feel that the well-mixed nature of the lake alleviates concerns about spatial disconnect among the sites.

3) I believe that more specific analysis is needed to better dig the time series data, especially the water clarity, as it seems that the water clarity did increase following the invasions. A time-series linear model may help uncover this trend;

We used a hierarchical generalized additive model to decompose the water clarity time series data (Figure 3). This method allowed us to parcel out non-linear seasonal (day-of-year), interannual, and spatial variability. The results of that analysis found a significant seasonal-invasion status interaction but not an overall trend related to invasion status.

4) The authors should dig deeper into their results. For example, the water flea density was not very high until 2017-18, and what caused this and the potential impacts on the compensatory relationship between zebra mussel and water flea should be discussed;

The spiny water flea densities in 2017 and 2018 were driven by high mean densities on 3 dates between the two years. When we calculated *Bythotrephes* production (Figure 4), integrating biomass formation through time, there was no difference between 2017-2018 and several of the previous years, although production in 2017-2018 was higher than production in several years, as well. This suggests that the resource demand by *Bythotrephes* on herbivorous crustaceans was variable across the years, but there was no trend in their energetic demand, despite changes in its density (Figure 4). Natural fluctuation of this magnitude is expected in such short-lived, r-selected species. It is also possible that fluctuations in populations of fish predators of *Bythotrephes* released *Bythotrephes* from top-down control in later years of the time series, which we discuss further below.

5) Some interpretations seemed weak due to a lack of statistical results, and I suggest linking discussion more tightly with results.

We added statistical analyses and referred to the figures more clearly throughout the text compared to the previous version.

L44: The impact of *Dreissena* increases rather than reduces the water clarity.

We corrected that error in L55.

L55: It would be very helpful if the mixed effects on phytoplankton being briefly summarized here.

We added, “with algae both increasing and not changing in invaded lakes” to the text in L66-67.

L98: Any evidence that the lake was not stratified?

We provided evidence in the form of heat maps from 2013-2019 produced from temperature profile data collected throughout the open water seasons of those years (Figure 2). Additionally, we referenced work from Fang et al. (2015) and Jacobson et al. (2008) that refers to the geometry of the lake basin in the context of water column stability and observations that while the lake may stratify in the open water season, that it is short-lived and polymictic (L48-50).

L137-139: I think this interpretation is very weak. If we focus on the period of 2008-2018, during which the yellow perch declined, it seems that the body size of zooplankton did not change over time. The density of water flea was medium to low during 2010-2016, during which the yellow perch and cisco were decline as well. Why?

We predicted that the size structure of the native zooplankton community would shift due to predation by *Bythotrephes*, and when we considered the entire zooplankton time series (2006-2018), mean size of native zooplankton increased (Mann Kendall test, $p=0.044$). After *Bythotrephes* was established (2010), mean size of the native zooplankton community was variable. In reality, the drivers of mean zooplankton size are likely much more complicated than we originally stated, and it is possible that a decrease in predation by yellow perch, which prefer large zooplankton, released the native zooplankton from fish predation enough to influence the size of the native zooplankton community. It is also possible that variability in the abundance of planktivorous fishes explains some of the variability we see in the

native zooplankton size structure, and in our dataset, nearshore yellow perch CPU was correlated with mean native zooplankton size ($r = -0.53$, $p=0.061$). Using the data we have, we are not able to discern the contribution of *Bythotrephes* and fish as drivers of native zooplankton size, but we agree both mechanisms are likely playing a role in structuring the size of the native zooplankton community and added that to the text (L 173-174).

Providing depth contour of the lake will be very helpful.

We updated the map (Figure 1) to include depth contour lines and the location of the water clarity measurement stations.

According to Fig. 1, the surveys of mussels were largely restrained to nearshore – possible shallow region – while the zooplankton and Chla survey spread out across the lake. This raises a question: the decoupling of the zebra mussel and zooplankton might underestimate the impacts of zebra mussels on Chla or overestimate the compensatory effect of the water flea on Chla. The authors should justify their selection or re-calculate the effect of mussels by focusing on the coupled sites (i.e., zebra mussel vs. Chla) only.

Based on diving surveys, only one of the zooplankton sampling stations is known to have zebra mussels present. Lake Mille Lacs is well-mixed, so we feel comfortable comparing filtering rates at the decoupled zebra mussel and zooplankton stations.

I suggest the authors combining the time series of some subpanels in Fig. 2 and Fig. 4 into a single panel for better comparisons among trophic levels.

We made the suggested change to the figures.

Reviewer #2 (Remarks to the Author):

Review of COMMSBIO-21-0240-T

Brief summary

The manuscript reports on a study that takes advantage of the co-invasion of a lake in Minnesota by two aquatic invaders– the spiny water flea (*Bythotrephes longimanus*) and the zebra mussel (*Dreissena polymorpha*) – to examine the co-effects of multiple invasive species on a native ecosystem. This was done using an annual, long-term observational monitoring program of the lake’s food web. The manuscript reports that water clarity in the lake remained stable after the establishment of *Bythotrephes* and *Dreissena*, which is surprising given that *Dreissena*-invaded waters typically increase in water clarity due to *Dreissena*’s filtering activities. The authors conclude that the maintenance of water clarity can best be explained by synchronous but opposite effects of *Bythotrephes* and *Dreissena* on food web processes, which they test by comparing modeled filtering rates of the native zooplankton versus *Dreissena*.

Overall impression

Broadly and conceptually, this is an interesting case study on the effects of multiple invasive species

on ecosystem processes. More specifically, the reported findings could have important applications as the two invasive species in the study are two of north America's most widespread and problematic. The long-term dataset (13 years) is a strength as there are several years sampled before and after the invasion events. Finally, the manuscript is generally well-written and well-referenced. However, like all 'natural experiments' the study does have some limitations. Further, the background information provided is quite dismissive of important foundational and ongoing work on multi-species invasions and their effects on ecosystems. The manuscript could be strengthened by (1) couching the work in a more balanced and accurate framework of multi-species invasions, and (2) better acknowledging the limitations of unreplicated or uncontrolled 'before-after' natural experiments.

Specific comments

(1) There has been much conceptual, methodological, and empirical work done on multi-species invasions – dating back to Simberloff and Von Holle's classic 'Invasional meltdown' paper in *Biological Invasions* in 1999. Therefore, I found lines 28-32 on Page 2 a bit dismissive of this body of work, and a somewhat inaccurate characterization of the field. There has even been a meta-analysis on multi-species invasions in the last five years (Jackson 2015 *Ecology*). See also the experimental test and conceptual framework by Braga et al. 2019 *Hydrobiologia*; the methodological and experimental work by Pearson et al. 2016 *Ecological Applications*; and empirical work on multi-species invasions by e.g., Adams et al. 2003 *Ecology Letters* and Montgomery et al. 2012 *Biological Invasions*.

We added text (L31-37) and citations (5-11) to provide a more thorough picture on literature related to multi-species invasions. Our lack of citations in the previous version was not meant to dismiss the existing work, but was a product of the word limit and limits on the number of citations when originally submitting the manuscript to *Nature*.

(2) Whereas long-term monitoring datasets enabling us to detect anthropogenic perturbations and their effects on natural systems are critical and all-too-rare, they are limited in their own right. I applaud the monitoring work done here and appreciate the amount of work that went into this study and think we gain some real insight from it. However, I do think the manuscript could be strengthened by acknowledging that some caution should be taken when using, or interpreting results from, a BACI-type design when no control system was available (i.e., Before-After only as you have here). See e.g., critical work by Underwood on this topic: *Ecological Applications* 1994, *Marine and Freshwater Research* 1991, etc., as well as that by Benedetti-Cecchi 2001 *Ecological Applications*, and Stewart-Oaten et al. 1986 *Ecology*).

In the Discussion (L212-214) we included an acknowledgement of the statistical issue related to our approach, of having before-after data from one impacted site. We compare our results to published effects of zebra mussels from other systems, acting as a control, of sorts.

Reviewer #3 (Remarks to the Author):

The authors provide a relatively high-level analysis of potentially competing effects of two invasive species on water quality and lower trophic level indicators in Lake Mille Lacs, MN. The authors purport to demonstrate a rare stability of phytoplankton biomass and water clarity despite a

concurrent invasion of zebra mussels and the spiny water flea. The lack of change in chlorophyll a and water clarity is attributed to grazing control of phytoplankton by zebra mussels and a concurrent relaxation of phytoplankton due to predation on herbivorous zooplankton by the spiny water flea. While this is a reasonably novel construct from first principles and would be of broad relevance to invasion ecology and lake management, I do not find the evidence overly compelling.

First, I find the sole focus on the annual timescale obscures potential information that may well (or not) support the authors conclusions. For example, at a sub-annual scale, do patterns of chl_a/clarity covary with anticipated time periods of 1) production and settling of dreissenid veligers or 2) peak abundance of *Bythotrephes* (and presumed predation of herbivores)? Comparisons of annual means subsumes variability that may well be instructive with respect to the claims made here.

Using a hierarchical generalized additive model, we examined clarity trends during the open water season (May-September) in Lake Mille Lacs, which allowed us to model seasonal trends, as well as effects of invasive species (Figure 3, Supplemental Figure 1). The interaction of day-of-year and invasion status (which AIS are present and when) supports our hypothesis that *Bythotrephes* is offsetting the effects of zebra mussels on water clarity. We first observe *Bythotrephes* in the zooplankton tows mid-May to early June each year, around day 180, when water clarity decreases, and we noted in the text (L 107-109) when we first observe *Bythotrephes* in our sample collection.

Second, the authors present a well supported argument that *Bythotrephes* has likely relaxed some grazing pressure on phytoplankton as all reported zooplankton metrics are consistent with known impacts of *Bythotrephes* invasion. However, they also argue that they can rule out changes in phytoplankton growth yet do not present information on primary production rates or other proxies (13C data? C:Chl ratios?) needed to evaluate it. The presence of periodic blooms does little to inform on growth rates of edible phytoplankton species. A passing reference is made to possible changes in non-point source nutrient inputs as a potential buffer to mussel grazing, but only present point in time measures of chl_a as evidence this is not occurring. Such reasoning ignores hydraulic forcings, circulation and wind and waves as possible reasons why one might also find elevated chl_a in some places. The authors also argue that water chemistry data collected at an outflow site that only partially overlaps the study period is somehow representative of nutrient inputs to lake infested with zebra mussels, which is a significant stretch and at odds with the general body of evidence relating mussel impacts on nutrient stores in lakes. This part of the hypothesis unfortunately remains speculative in my opinion without more robust data.

Unfortunately, neither isotope data nor carbon:chlorophyll data exist for this lake system. We also do not have data for other methodologies to assess primary production rates. We noted in the text (L146-147) that it is possible primary production increased without a change in chlorophyll concentrations. We secured additional total phosphorus data from the lake (2000-2018) and we included those data in the manuscript and removed the water chemistry data from a downstream monitoring station. A trend analysis of the total phosphorus dataset failed to show a trend in phosphorus from 2000-2018 (Figure 4H, $p > 0.05$) that would support changes to productivity. Likewise, lake temperature data were not different among to invasion status groups, driving increased growth and production (Supplementary Figure 3). Additionally, we added data about land use to support an argument that nutrient inputs to the lake have not increased via watershed disturbance during the study period

Lastly, some of this data appears in a recently published paper (Jones and Montz 2020; Biol. Inv. 9(4):772-792) where those authors have come to similar conclusions and raised the idea that *Bythotrephes* may be responsible for the apparent lack of change in clarity and chl_a. I would recommend the authors acknowledge that work should they decide to revise the paper.

We added a citation in the text to reflect the conclusion in Jones and Montz (2020) at Line 133.

Specific comments

Statistical methods – While I can understand the bootstrapping approach used here, I wondered if a mixed effect modeling framework might provide additional insight into the sequence of changes and potential confounders. It can easily be done from a pre/post perspective or change over time. Further, I wonder if bootstrapping across data derived from one source (zooplankton) vs others which appear to come from multiple sources (i.e. secchi disc) is appropriate. A mixed model framework can handle that aspect fairly easily, while still providing estimates of the main effects. It appears Jones and Montz (2020) Biol. Inv. 9(4):772-792 have already taken this approach for some of this data with respect to mussel effects on water clarity and chl_a.

We used a hierarchical generalized additive model to account for seasonal patterns (day-of-year), as well as the presence of zebra mussels or zebra mussels and spiny water flea. We also included an AIS*day-of-year interaction, as the plotted data suggested that there were seasonal differences among the different assemblages of invasive species. We included random effects for year and monitoring location.

Line 33 – can you confirm only *D.polymorpha* present (i.e. no *D.bugensis*)?

Yes, currently in Minnesota, *D. bugensis* is only found in the Mississippi River and Lake Superior.

Line 71 – typical time to peak population of *D.polymorpha* in NA is ~ 2.5 yrs – why the long lag here? Possible that population undersampled?

D. polymorpha was first found in Lake Mille Lakes by MNDNR Fisheries divers who were assessing fish habitat in 2005 and happened upon an individual mussel. That discovery led to an intensive effort by MNDNR at 60 stations, resulting in the observation of 4 individual mussels. The first report of a zebra mussel from the public was in 2009, from a dock. Without the MNDNR diving effort, zebra mussels likely would not have been detected until then, which is 3 years before the population peaked in 2012.

Line 113 – ruling out bottom – up processes would probably entail either modeled primary production or some measure of primary production no? blooms (depending on species) may not necessarily represent edible algae, and they tend to represent a small time frame of a typical season.

We agree that standing stock biomass based on chlorophyll-a does not present the entire picture, we provided additional evidence to the lack of drivers of increased primary production (no change in TP

through time, no difference in water temperatures among the three invasion status groups, no change in land cover).

Lines 121-125 – why use chla at point sources to infer nutrient loading has not increased? Would not actual load estimates be a better proxy? Furthermore, one assumes that water still circulates within the lake so point in time chla samples really don't provide convincing evidence one way or the other.

No nutrient data exist for the inlet streams to the lake to calculate load, but we obtained total phosphorus data from the duration of the study, which we included (Figure 4H). We added text to acknowledge the possibility that the data we presented may not capture increases in nutrient loads (L149-151). Additionally, we quantified land cover in the Mille Lacs watershed using the National Land Cover Database in 2001, 2006, 2011, and 2016 and found no change in land cover during that time (see Supplemental Information). A study of 1330 Minnesota lakes by Cross and Jacobson (2013) found that increased anthropogenic disturbance (agricultural and urban development) and maximum water depth were the most influential characteristics of a lake that drive total phosphorus concentrations, which we cited in the text.

Lines 141-142 – are there other data to support or refute the clarity hypotheses? i.e. no change in macrophyte cover? Extent? Depth of colonization? Simon and Perleberg (2015) reported no change in maximum roo

The plant community story as it relates to zebra mussels in the lake is complex. Simon et al. (2020) reviewed the data from macrophyte surveys in the lake in 2009, 2010, 2014, 2017, and 2019. They found fewer plants in deeper water in the recent surveys in Lake Mille Lacs and attribute that to zebra mussel attachment on the plants themselves. Additionally, the deeper habitat in the lake is poor for macrophytes, due to the sandy substrata.

Lines 125-128 - The time frame of water quality at an outlet site between 1955 – 2010 tells me little about trends in nutrients entering the study lake from 2006-2018. A decline in outflow concentrations is not unexpected post colonization by mussels.

We understand the problems related to using the nutrient data from a downstream stream monitoring station. We removed the data from the outlet stream. We obtained total phosphorus data for the time series and included those data in the manuscript (Figure 4H).

Line 329 – why not include GDD and solar insolation as other proxies of energy flux – such variables are known to produce coherent patterns in biological response variables (i.e. chla) – could these also explain why the signal in chla is weak?

We compared water temperature (cumulative degree days, base 0 °C) for the among the three time periods. There was no difference between the groups at day-of-year 270, which represents the end of our seasonal sampling period (L158-160, Supplementary Figure 3).

Line 336 -338 – Chla can still vary vertically even in well mixed systems, particularly during quiescent periods. Do you have additional information to support this? Perhaps fluorometer data? Chla data at different depths? Further; were samples extracted using the same solvent across the time series?

We do not have chlorophyll-*a* data from discrete depths or from profiles. The Minnesota Department of Agriculture Laboratory analyzed all the chlorophyll samples and used buffered acetone as a solvent, following EPA Method 446.0 (Arar, 1997).

Lines 347-349 - The exclusion of nearshore chla collected near point sources is somewhat confusing – I get the desire to remove confounders, but it is not like dreissenids eschew a free meal – perhaps a note to say that grazing rates could be underestimated by doing this?

The grazing rates are not based on chlorophyll data but on population biomass-weighted water column clearance rates derived from the zebra mussel population in the western basin of Lake Erie (Zhang et al 2011). We estimated population biomass using the mean mass from Jones and Montz, extrapolating the biomass/m² to the proportion of the lake bottom covered with zebra mussels (km², Jones and Montz). Error in the estimate of zebra mussel grazing rates are related to error in estimating densities in the mussels. We included all chlorophyll-*a* data in the most recent version of the analyses (Figure 4) and removed the supplementary figure which separated the nearshore sampling locations from the rest of the monitoring stations.

Fig 2. It appears as if there is a slight upward trend in clarity but nothing obvious in chla. If grazing is indeed best placed to explain the lack of pattern, could a shift to smaller cells account for a small increase in clarity while keeping chla the same?

The results of the hierarchical GAM do not indicate an overall increase in water clarity in the lake (Figure 3). The model does identify increased water clarity in the early summer (until about day 180) since 2010, compared to the 2000-2005 and 2006-2009 data. The period of increased water clarity ceases at the end of June, around the time when *Bythotrephes* is first collected in the zooplankton tows. It is true that a shift in the taxonomy of the algal community could vary shading while keeping the water column chlorophyll the same. However, we did not see an increase in *Daphnia*, which we would expect if there was an increase in the number of small algal cells. Unfortunately, we do not have phytoplankton community data from the lake to investigate changes in community structure.

Reviewers' comments:

Reviewer #1 (Remarks to the Author):

The authors report a 13-yr survey on two notorious invaders in North American freshwater ecosystems – zebra mussels and spiny water flea – in Lake Mille Lacs in Minnesota, US. This is a rare case study that covers both before and after the invasion of the two invaders, providing an important model to investigate the impacts of co-invasion on invaded ecosystems. The author concluded that the surprisingly unchanged water clarity and summer phytoplankton production were a result of competing effects of ZM and SWF. Specifically, a relaxation in predation pressure on phytoplankton by reduced zooplankton (due to consumption of SWF) offset the depletion effect of ZM on phytoplankton. I have read the revised paper thoroughly, and I applaud the authors' efforts in revising this work, which has improved a lot. While I think this work is interesting and important enough for publication, I am still not fully convinced by the current presentation. The revised version seems still to be built too much on speculation rather than direct and solid evidence that supports the main conclusions, even though I believe that such speculation is somewhat reliable. I have a few suggestions to improve this work:

Given the short-lived fact of the zooplankton, the authors should consider finer temporal scales (e.g., monthly –semimonthly data) in illustrating the population dynamics of the spiny water flea and other herbivorous zooplankton groups to directly test and support the bottom-up control hypothesis that underlies the paper. Similarly, top-down control should also be validated using the fish and zooplankton data.

While the opposite effects of the two invaders on water clarity are predictable, it is confounding to attribute the unchanged temporal pattern to the relaxed predation pressure by zooplankton even though potential contributors/confounders have been discussed. The Lake Mille Lacs is a large, shallow lake where resuspension of sediment and nutrient is possible. It is well-established in the literature that the impacts of dreissenid mussels on water clarity and primary production are location (e.g., nearshore vs. offshore) –dependent, but this was not reflected in the current version even though the depth of sampled sites varied substantially. I understand the nature of lacking controls for such a long-term survey dataset, and I encourage the authors to take advantage of such information to rule out confounders by looking up sites with vs. without zebra mussels.

Reviewer #1:

1) Given the short-lived fact of the zooplankton, the authors should consider finer temporal scales (e.g., monthly –semimonthly data) in illustrating the population dynamics of the spiny water flea and other herbivorous zooplankton groups to directly test and support the bottom-up control hypothesis that underlies the paper. Similarly, top-down control should also be validated using the fish and zooplankton data.

We plotted mean daily *Bythotrephes* densities versus densities for three groups of native zooplankton (cladocerans, calanoids, and cyclopoids, Figure 5), which is a better representation of shorter timescale population dynamics between *Bythotrephes* and its prey. We also plotted mean gill net catches of the two main planktivorous fishes in Lake Mille Lacs (yellow perch and cisco) versus the mean annual densities of the native zooplankton. We weren't able to further decompose the fish-zooplankton relationships temporally, as the fish were sampled once during each year.

There is evidence of strong top-down control on native zooplankton by the non-native zooplankton *Bythotrephes*, similar to other comparisons of these zooplankton groups in the Laurentian Great Lakes (Lehman and Cáceres 1993). The strong non-linear relationship between the native zooplankton populations and *Bythotrephes* is predicted by theoretical ecology and supported by a metaanalysis of relationships between invasive non-native and native species (Bradley et al 2019).

The relationships between native zooplankton and the fish predators do not provide evidence of top-down control. There is, however, evidence of bottom-up control on yellow perch based on the correlation from the nearshore gill net data and both the cladoceran and cyclopid zooplankton ($r = 0.722$ and 0.799 , respectively, $p < 0.01$ for both).

2) While the opposite effects of the two invaders on water clarity are predictable, it is confounding to attribute the unchanged temporal pattern to the relaxed predation pressure by zooplankton even though potential contributors/confounders have been discussed. The Lake Mille Lacs is a large, shallow lake where resuspension of sediment and nutrient is possible. It is well-established in the literature that the impacts of dreissenid mussels on water clarity and primary production are location (e.g., nearshore vs. offshore) – dependent, but this was not reflected in the current version even though the depth of sampled sites varied substantially. I understand the nature of lacking controls for such a long-term survey dataset, and I encourage the authors to take advantage of such information to rule out confounders by looking up sites with vs. without zebra mussels.

We addressed this comment by incorporating a couple changes to the analyses. First, we added clarity measurements from the MNDNR dataset that we were preciously unaware of. These measurements were taken at the zooplankton/water chemistry monitoring stations on days water and plankton were collected. Including this dataset added 792 observations in our generalized additive model and

increased the spatial distribution of observations. To compare clarity readings at all of the locations of data collection in the lake, we simulated clarity at each location, holding other variables constant. We plotted simulated data versus multiple spatial metrics (distance from nearest known zebra mussel bed, distance from shore, and water depth) to see if there are spatial impacts of zebra mussels on water clarity in the lake. There were weak, non-significant correlations between spatial metrics and water clarity (Supplementary Figure 2). We used simulated clarity data for the spatial comparisons because of the known impacts of covariates on clarity and the lack of synchrony in the collection of clarity data among the stations. We updated the map and all figures with clarity data, as well as re-running the generalized additive model, to include observations from the additional clarity dataset.

Bradley, B. A. et al. Disentangling the abundance-impact relationship for invasive species. *Proc. Natl. Acad. Sci. U.S.A.* **116**, 9919-9924 (2019).

REVIEWERS' COMMENTS:

Reviewer #1 (Remarks to the Author):

I applaud the authors' efforts in revising this work. The revised paper has addressed my concerns in the previous version.

Minor comments:

L45: specify the densities.

L592: should be black cross, not triangles.

Fig. 1: The number of stations on the map does not match the figure caption.